# Robust Spatial Filtering with Beta Divergence

**Wojciech Samek**[1,4]    **Duncan Blythe**[1,4]    **Klaus-Robert Müller**[1,2]    **Motoaki Kawanabe**[3]

[1]Machine Learning Group, Berlin Institute of Technology (TU Berlin), Berlin, German
[2]Department of Brain and Cognitive Engineering, Korea University, Seoul, Korea
[3]ATR Brain Information Communication Research Laboratory Group, Kyoto, Japan
[4]Bernstein Center for Computational Neuroscience, Berlin, Germany

## Abstract

The efficiency of Brain-Computer Interfaces (BCI) largely depends upon a reliable extraction of informative features from the high-dimensional EEG signal. A crucial step in this protocol is the computation of spatial filters. The Common Spatial Patterns (CSP) algorithm computes filters that maximize the difference in band power between two conditions, thus it is tailored to extract the relevant information in motor imagery experiments. However, CSP is highly sensitive to artifacts in the EEG data, i.e. few outliers may alter the estimate drastically and decrease classification performance. Inspired by concepts from the field of information geometry we propose a novel approach for robustifying CSP. More precisely, we formulate CSP as a divergence maximization problem and utilize the property of a particular type of divergence, namely beta divergence, for robustifying the estimation of spatial filters in the presence of artifacts in the data. We demonstrate the usefulness of our method on toy data and on EEG recordings from 80 subjects.

## 1   Introduction

Spatial filtering is a crucial step in the reliable decoding of user intention in Brain-Computer Interfacing (BCI) [1, 2]. It reduces the adverse effects of volume conduction and simplifies the classification problem by increasing the signal-to-noise-ratio. The Common Spatial Patterns (CSP) [3, 4, 5, 6] method is one of the most widely used algorithms for computing spatial filters in motor imagery experiments. A spatial filter computed with CSP maximizes the differences in band power between two conditions, thus it aims to enhance detection of the synchronization and desynchronization effects occurring over different locations of the sensorimotor cortex after performing motor imagery. It is well known that CSP may provide poor results when artifacts are present in the data or when the data is non-stationary [7, 8]. Note that artifacts in the data are often unavoidable and can not always be removed by preprocessing, e.g. with Independent Component Analysis. They may be due to eye movements, muscle movements, loose electrodes, sudden changes of attention, circulation, respiration, external events, among the many possibilities. A straight forward way to robustify CSP against overfitting is to regularize the filters or the covariance matrix estimation [3, 7, 9, 10, 11]. Several other strategies have been proposed for estimating spatial filters under non-stationarity [12, 8, 13, 14].

In this work we propose a novel approach for robustifying CSP inspired from recent results in the field of information geometry [15, 16]. We show that CSP may be formulated as a divergence maximization problem, in particular we prove by using Cauchy's interlacing theorem [17] that the spatial filters found by CSP span a subspace with maximum symmetric Kullback-Leibler divergence between the distributions of both classes. In order to robustify the CSP algorithm against the influence of outliers we propose solving the divergence maximization problem with a particular type of

divergence, namely beta divergence. This divergence has been successfully used for robustifying algorithms such as Independent Component Analysis (ICA) [18] and Non-negative Matrix Factorization (NMF) [19]. In order to capture artifacts on a trial-by-trial basis we reformulate the CSP problem as sum of trial-wise divergences and show that our method downweights the influence of artifactual trials, thus it robustly integrates information from all trials.

The remainder of this paper is organized as follows. Section 2 introduces the divergence-based framework for CSP. Section 3 describes the beta-divergence CSP method and discusses its robustness property. Section 4 evaluates the method on toy data and EEG recordings from 80 subjects and interprets the performance improvement. Section 5 concludes the paper with a discussion. An implementation of our method is available at `http://www.divergence-methods.org`.

## 2 Divergence-Based Framework for CSP

Spatial filters computed by the Common Spatial Patterns (CSP) [3, 4, 5] algorithm have been widely used in Brain-Computer Interfacing as they are well suited to discriminate between distinct motor imagery patterns. A CSP spatial filter $\mathbf{w}$ maximizes the variance of band-pass filtered EEG signals in one condition while minimizing it in the other condition. Mathematically the CSP solution can be obtained by solving the generalized eigenvalue problem

$$\mathbf{\Sigma}_1 \mathbf{w}_i = \lambda_i \mathbf{\Sigma}_2 \mathbf{w}_i \tag{1}$$

where $\mathbf{\Sigma}_1$ and $\mathbf{\Sigma}_2$ are the estimated (average) $D \times D$ covariance matrices of class 1 and 2, respectively. Note that the spatial filters $\mathbf{W} = [\mathbf{w}_1 \ldots \mathbf{w}_D]$ can be sorted by importance $\alpha_1 = \max\{\lambda_1, \frac{1}{\lambda_1}\} > \ldots > \alpha_D = \max\{\lambda_D, \frac{1}{\lambda_D}\}$.

### 2.1 divCSP Algorithm

Information geometry [15] has provided useful frameworks for developing various machine learning (ML) algorithms, e.g. by optimizing divergences between two different probability distributions [20] [21]. In particular, a series of robust ML methods have been successfully obtained from Bregman divergences which are generalization of the Kullback-Leibler (KL) divergence [22]. Among them, we employ in this work the beta divergence. Before proposing our novel algorithm, we show that CSP can also be interpreted as maximization of the symmetric KL divergence.

**Theorem 1**: Let $\mathbf{W} = [\mathbf{w}_1 \ldots \mathbf{w}_d]$ be the $d$ top (sorted by $\alpha_i$) spatial filters computed by CSP and let $\mathbf{\Sigma}_1$ and $\mathbf{\Sigma}_2$ denote the covariance matrices of class 1 and 2. Let $\mathbf{V}^\top = \tilde{\mathbf{R}}\mathbf{P}$ be a $d \times D$ dimensional matrix that can be decomposed into a whitening projection $\mathbf{P} \in \mathbb{R}^{D \times D}$ ($\mathbf{P}(\mathbf{\Sigma}_1 + \mathbf{\Sigma}_2)\mathbf{P}^\top = \mathbf{I}$) and an orthogonal projection $\tilde{\mathbf{R}} \in \mathbb{R}^{d \times D}$. Then

$$\mathrm{span}(\mathbf{W}) = \mathrm{span}(\mathbf{V}^*) \tag{2}$$

$$\text{with } \mathbf{V}^* = \underset{\mathbf{V}}{\mathrm{argmax}} \, \tilde{D}_{kl}\left(\mathbf{V}^\top \mathbf{\Sigma}_1 \mathbf{V} \, \| \, \mathbf{V}^\top \mathbf{\Sigma}_2 \mathbf{V}\right) \tag{3}$$

where $\tilde{D}_{kl}(\cdot \, \| \, \cdot)$ denotes the symmetric Kullback-Leibler Divergence[1] between zero mean Gaussians and $\mathrm{span}(\mathbf{M})$ stands for the subspace spanned by the columns of matrix $\mathbf{M}$. Note that [23] has provided a proof for the special case of one spatial filter, i.e. for $\mathbf{V} \in \mathbb{R}^{D \times 1}$.

**Proof**: See appendix and supplement material.

The objective function that is maximized in Eq. (3) can be written as

$$\mathcal{L}_{kl}(\mathbf{V}) = \frac{1}{2}\mathrm{tr}\left((\mathbf{V}^\top \mathbf{\Sigma}_1 \mathbf{V})^{-1}(\mathbf{V}^\top \mathbf{\Sigma}_2 \mathbf{V})\right) + \frac{1}{2}\mathrm{tr}\left((\mathbf{V}^\top \mathbf{\Sigma}_2 \mathbf{V})^{-1}(\mathbf{V}^\top \mathbf{\Sigma}_1 \mathbf{V})\right) - \mathrm{d}. \tag{4}$$

In order to cater for artifacts on a trial-by-trial basis we need to reformulate the above objective function. Instead of maximizing the divergence between the average class distributions we propose to optimize the sum of trial-wise divergences

$$\mathcal{L}_{sumkl}(\mathbf{V}) = \sum_{i=1}^{N} \tilde{D}_{kl}\left(\mathbf{V}^\top \mathbf{\Sigma}_1^i \mathbf{V} \, \| \, \mathbf{V}^\top \mathbf{\Sigma}_2^i \mathbf{V}\right), \tag{5}$$

where $\boldsymbol{\Sigma}_1^i$ and $\boldsymbol{\Sigma}_2^i$ denote the covariance matrices estimated from the $i$-th trial of class 1 and class 2, respectively, and $N$ is the number of trials per class. Note that the reformulated problem is not equivalent to CSP; in Eq. (4) averaging is performed w.r.t. the covariance matrices, whereas in Eq. (5) it is performed w.r.t. the divergences. We denote the former approach by $kl$-divCSP and the latter one by sum$kl$-divCSP. The following theorem relates both approaches in the asymptotic case.

**Theorem 2**: Suppose that the number of discriminative sources is one; then let $c$ be such that $D/n \to c$ as $D, n \to \infty$ ($D$ dimensions, $n$ data points per trial). Then if there exists $\gamma(c)$ with $N/D \to \gamma(c)$ for $N \to \infty$ ($N$ the number of trials) then the empirical maximizer of $\mathcal{L}_{sumkl}(\mathbf{v})$ (and of course also of $\mathcal{L}_{kl}(\mathbf{v})$) converges almost surely to the true solution.

**Sketched Proof**: See appendix.

Thus Theorem 2 says that both divergence-based CSP variants $kl$-divCSP and sum$kl$-divCSP almost surely converge to the same (true) solution in the asymptotic case. The theorem can be easily extended to multiple discriminative sources.

## 2.2 Optimization Framework

We use the methods developed in [24], [25] and [26] for solving the maximization problems in Eq. (4) and Eq. (5). The projection $\mathbf{V} \in \mathbb{R}^{D \times d}$ to the $d$-dimensional subspace can be decomposed into three parts, namely $\mathbf{V}^\top = \mathbf{I}_d \mathbf{R} \mathbf{P}$ where $\mathbf{I}_d$ is an identity matrix truncated to the first $d$ rows, $\mathbf{R}$ is a rotation matrix with $\mathbf{R}\mathbf{R}^\top = \mathbf{I}$ and $\mathbf{P}$ is a whitening matrix. The optimization process consists of finding the rotation $\mathbf{R}$ that maximizes our objective function and can be performed by gradient descent on the manifold of orthogonal matrices. More precisely, we start with an orthogonal matrix $\mathbf{R}_0$ and find an orthogonal update $\mathbf{U}$ in the $k$-th step such that $\mathbf{R}_{k+1} = \mathbf{U}\mathbf{R}_k$. The update matrix is chosen by identifying the direction of steepest descent in the set of orthogonal transformations and then performing a line search along this direction to find the optimal step. Since the basis of the extracted subspace is arbitrary (one can right multiply a rotation matrix to $\mathbf{V}$ without changing the divergence), we select the principal axes of the data distribution of one class (after projection) as basis in order to maximally separate the two classes. The optimization process is summarized in Algorithm 1 and explained in the supplement material of the paper.

---

**Algorithm 1** Divergence-based Framework for CSP

1: **function** DIVCSP($\boldsymbol{\Sigma}_1, \boldsymbol{\Sigma}_2, d$)
2:      Compute the whitening matrix $\mathbf{P} = \boldsymbol{\Sigma}^{-\frac{1}{2}}$
3:      Initialise $\mathbf{R}_0$ with a random rotation matrix
4:      Whiten and rotate the data $\boldsymbol{\Sigma}_c = (\mathbf{R}_0\mathbf{P})\boldsymbol{\Sigma}_c(\mathbf{R}_0\mathbf{P})^\top$ with $c = \{1, 2\}$
5:      **repeat**
6:          Compute the gradient matrix and determine the step size (see supplement material)
7:          Update the rotation matrix $\mathbf{R}_{k+1} = \mathbf{U}\mathbf{R}_k$
8:          Apply the rotation to the data $\boldsymbol{\Sigma}_c = \mathbf{U}\boldsymbol{\Sigma}_c\mathbf{U}^\top$
9:      **until** convergence
10:      Let $\mathbf{V}^\top = \mathbf{I}_d\mathbf{R}_{k+1}\mathbf{P}$
11:      Rotate $\mathbf{V}$ by $\mathbf{G} \in \mathbb{R}^{d \times d}$ where $\mathbf{G}$ are eigenvectors of $\mathbf{V}^\top\boldsymbol{\Sigma}_1\mathbf{V}$
12:      **return** $\mathbf{V}$
13: **end function**

---

## 3 Beta Divergence CSP

Robustness is a desirable property of algorithms that work in data setups which are known to be contaminated by outliers. For example, in the biomedical fields, signals such as EEG may be highly affected by artifacts, i.e. outliers, which may drastically influence statistical estimation. Note that both of the above approaches $kl$-divCSP and sum$kl$-divCSP are not robust w.r.t. artifacts as they both perform simple (non-robust) averaging of the covariance matrices and of the divergence terms, respectively. In this section we show that by using beta divergence we robustify the averaging of the divergence terms as beta divergence downweights the influence of outlier trials.

Beta divergence was proposed in [16, 27] and is defined (for $\beta > 0$) as

$$D_\beta\left(f(x) \;\|\; g(x)\right) = \frac{1}{\beta}\int (f^\beta(x) - g^\beta(x))f(x)dx - \frac{1}{\beta+1}\int (f^{\beta+1}(x) - g^{\beta+1}(x))dx, \quad (6)$$

where $f(x)$ and $g(x)$ are two probability distributions. Like every statistical divergence it is always positive and equals zero iff $g = f$ [15]. The symmetric version of beta divergence $\tilde{D}_\beta(f(x) \;\|\; g(x)) = D_\beta(f(x) \;\|\; g(x)) + D_\beta(g(x) \;\|\; f(x))$ can be interpreted as discrepancy between two probability distributions. One can show easily that beta and Kullback-Leibler divergence coincide as $\beta \to 0$.

In the context of parameter estimation, one can show that minimizing the divergence function from an empirical distribution $p$ to the statistical model $q(\phi)$ is equivalent to maximizing the $\Psi$-likelihood $\bar{L}_{\Psi_\beta}(\phi)$

$$\operatorname*{argmin}_{q(\phi)} D_\beta(p \,\|\, q(\phi)) = \operatorname*{argmax}_{q(\phi)} \bar{L}_{\Psi_\beta}(q(\phi)) \qquad (7)$$

$$\text{with } \bar{L}_{\Psi_\beta}(q(\phi)) = \frac{1}{n}\sum_{i=1}^{n} \Psi_\beta(\ell(x_i, q(\phi))) - b_{\Psi_\beta}(\phi) \;\text{ and }\; \Psi_\beta(z) = \frac{\exp(\beta z) - 1}{\beta}, \quad (8)$$

where $\ell(x_i, q(\phi))$ denotes the log-likelihood of observation $x_i$ and distribution $q(\phi)$, and $b_{\Psi_\beta}(\phi) := (\beta+1)^{-1}\int q(\phi)^{\beta+1}dx$. Basu et al. [27] showed that the $\Psi$-likelihood method weights each observation according to the magnitude of likelihood evaluated at the observation; if an observation is an outlier, i.e. of lower likelihood, then it is downweighted. Thus, beta divergence allows to construct robust estimators as samples with low likelihood are downweighted (see also M-estimators [28]).

## $\beta$-divCSP Algorithm

We propose applying beta divergence to the objective function in Eq. (5) in order to downweight the influence of artifacts in the computation of spatial filters. An overview over the different divergence-based CSP variants is provided in Figure 1. The objective function of our $\beta$-divCSP approach is

$$\mathcal{L}_\beta(\mathbf{V}) = \sum_i \tilde{D}_\beta\left(\mathbf{V}^T\boldsymbol{\Sigma}_1^i\mathbf{V} \;\|\; \mathbf{V}^T\boldsymbol{\Sigma}_2^i\mathbf{V}\right) \qquad (9)$$

$$= \frac{1}{\beta}\sum_i \left(\int g_i^{\beta+1}dx + \int f_i^{\beta+1}dx - \int f_i^\beta g_i dx - \int f_i g_i^\beta dx\right), \qquad (10)$$

with $f_i = \mathcal{N}\left(\mathbf{0}, \bar{\boldsymbol{\Sigma}}_1^i\right)$ and $g_i = \mathcal{N}\left(\mathbf{0}, \bar{\boldsymbol{\Sigma}}_2^i\right)$ being the zero-mean Gaussian distributions with projected covariances $\bar{\boldsymbol{\Sigma}}_1^i = \mathbf{V}^T\boldsymbol{\Sigma}_1^i\mathbf{V} \in \mathbb{R}^{d\times d}$ and $\bar{\boldsymbol{\Sigma}}_2^i = \mathbf{V}^T\boldsymbol{\Sigma}_2^i\mathbf{V} \in \mathbb{R}^{d\times d}$, respectively.

One can show easily (see the supplement file to this paper) that $\mathcal{L}_\beta(\mathbf{V})$ has an explicit form

$$\gamma\sum_i \left(|\bar{\boldsymbol{\Sigma}}_1^i|^{-\frac{\beta}{2}} + |\bar{\boldsymbol{\Sigma}}_2^i|^{-\frac{\beta}{2}} - (\beta+1)^{\frac{d}{2}}\left(|\bar{\boldsymbol{\Sigma}}_2^i|^{\frac{1-\beta}{2}}|\beta\bar{\boldsymbol{\Sigma}}_1^i + \bar{\boldsymbol{\Sigma}}_2^i|^{-\frac{1}{2}} + |\bar{\boldsymbol{\Sigma}}_1^i|^{\frac{1-\beta}{2}}|\beta\bar{\boldsymbol{\Sigma}}_2^i + \bar{\boldsymbol{\Sigma}}_1^i|^{-\frac{1}{2}}\right)\right), (11)$$

with $\gamma = \frac{1}{\beta}\sqrt{\frac{1}{(2\pi)^{\beta d}(\beta+1)^d}}$. We use Algorithm 1 to maximize the objective function of $\beta$-divCSP. In the following we show that the robustness property of $\beta$-divCSP can be directly understood from inspection of its objective function.

Assume $\bar{\boldsymbol{\Sigma}}_1^i$ and $\bar{\boldsymbol{\Sigma}}_2^i$ are full rank $d \times d$ covariance matrices. We investigate the behaviour of the objective functions of $\beta$-divCSP and sum$kl$-divCSP when $\bar{\boldsymbol{\Sigma}}_1^i$ is constant and $\bar{\boldsymbol{\Sigma}}_2^i$ becomes very large, e.g. because it is affected by artifacts. It is not hard to see that for $\beta > 0$ the objective function $\mathcal{L}_\beta$ does not go to infinity but is constant as $\bar{\boldsymbol{\Sigma}}_2^i$ becomes arbitrarily large. The first term of the objective function $|\bar{\boldsymbol{\Sigma}}_1^i|^{-\frac{\beta}{2}}$ is constant with respect to changes of $\bar{\boldsymbol{\Sigma}}_2^i$ and all the other terms go to zero as $\bar{\boldsymbol{\Sigma}}_2^i$ increases. Thus the influence function of the $\beta$-divCSP estimator is bounded w.r.t. changes in $\bar{\boldsymbol{\Sigma}}_2^i$ (the same argument holds for changes of $\bar{\boldsymbol{\Sigma}}_1^i$). Note that this robustness property vanishes when applying Kullback-Leibler divergences Eq. (4) as the trace term $\operatorname{tr}\left((\bar{\boldsymbol{\Sigma}}_1^i)^{-1}\bar{\boldsymbol{\Sigma}}_2^i\right)$ is not bounded when $\bar{\boldsymbol{\Sigma}}_2^i$ becomes arbitrarily large, thus this artifactual term will dominate the solution.

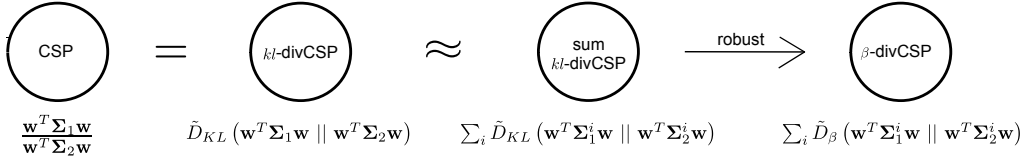

Figure 1: Relation between the different CSP formulations outlined in this paper.

## 4 Experimental Evaluation

### 4.1 Simulations

In order to investigate the effects of artifactual trials on CSP and $\beta$-divCSP we generate data $\mathbf{x}(t)$ using the following mixture model

$$\mathbf{x}(t) \quad = \quad A \begin{bmatrix} \mathbf{s}^{dis}(t) \\ \mathbf{s}^{ndis}(t) \end{bmatrix} + \epsilon, \tag{12}$$

where $A \in \mathbb{R}^{10 \times 10}$ is a random orthogonal mixing matrix, $\mathbf{s}^{dis}$ is a discriminative source sampled from a zero mean Gaussian with variance 1.8 in one condition and 0.2 in the other one, $\mathbf{s}^{ndis}$ are 9 sources with variance 1 in both conditions and $\epsilon$ is a noise variable with variance 2. We generate 100 trials per condition, each consisting of 200 data points. Furthermore we randomly add artifacts with variance 10 independently to each data dimension (i.e. virtual electrode) and trial with varying probability and evaluate the angle between the true filter extracting the source activity of $\mathbf{s}^{dis}$ and the spatial filter computed by CSP and $\beta$-divCSP. The median angles are shown in Figure 2 using 100 repetitions. One can clearly see that the angle error between the spatial filter extracted by CSP and the true one increases with larger artifact probability. Furthermore one can see from the figure that using very small $\beta$ values does not attenuate the artefact problem, but it rather increases the error by adding up trial-wise divergences without downweighting outliers. However, as the $\beta$ value increases the artifactual trials are downweighted and a robust average is computed over the trial-wise divergence terms. This increased robustness significantly reduces the angle error.

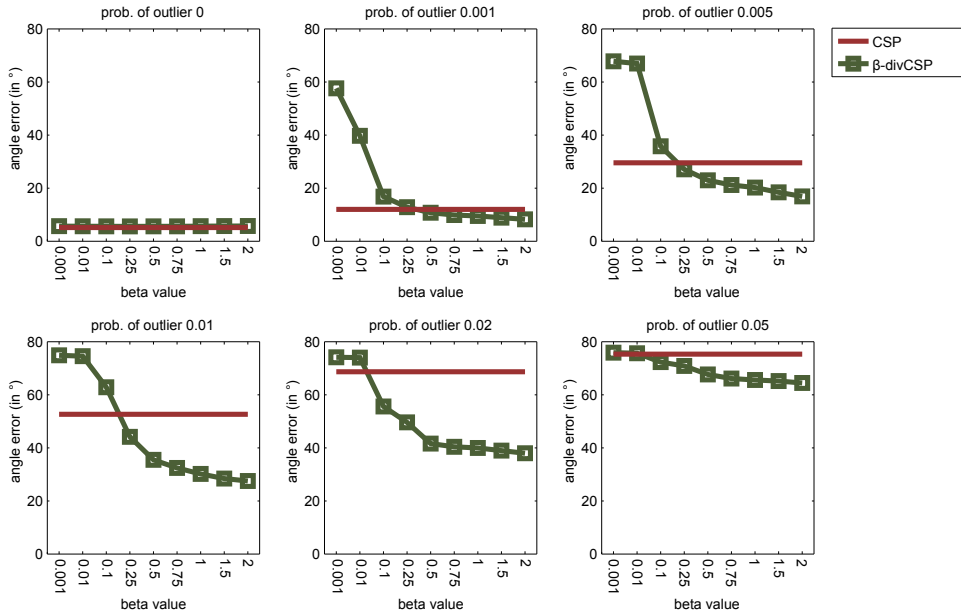

Figure 2: Angle between the true spatial filter and the filter computed by CSP and $\beta$-divCSP for different probabilities of artifacts. The robustness of our approach increases with the $\beta$ value and significantly outperforms the CSP solution.

## 4.2 Data Sets and Experimental Setup

The data set [29] used for the evaluation contains EEG recordings from 80 healthy BCI-inexperienced volunteers performing motor imagery tasks with the left and right hand or feet. The subjects performed motor imagery first in a calibration session and then in a feedback mode in which they were required to control a 1D cursor application. Activity was recorded from the scalp with multi-channel EEG amplifiers using 119 Ag/AgCl electrodes in an extended 10-20 system sampled at 1000 Hz (downsampled to 100 Hz) and a band-pass from 0.05 to 200 Hz. Three runs with 25 trials of each motor condition were recorded in the calibration session and the two best classes were selected; the subjects performed feedback with three runs of 100 trials. Both sessions were recorded on the same day.

For the offline analysis we manually select 62 electrodes densely covering the motor cortex, extract a time segment located from 750ms to 3500ms after the cue indicating the motor imagery class and filter the signal in 8-30 Hz using a 5-th order Butterworth filter. We do not apply manual or automatic rejection of trials or electrodes and use six spatial filters for feature extraction. For classification we apply Linear Discriminant Analysis (LDA) after computing the logarithm of the variance on the spatially filtered data. We measure performance as misclassification rate and normalize the covariance matrices by dividing them by their traces. The parameter $\beta$ is selected from the set of 15 candidates $\{0, 0.0001, 0.001, 0.01, 0.05, 0.1, 0.15, 0.2, 0.25, 0.5, 0.75, 1, 1.5, 2, 5\}$ by 5-fold cross-validation on the calibration data using minimal training error rate as selection criterion. For faster convergence we use the rotation part of the CSP solution as initial rotation matrix $\mathbf{R}_0$.

## 4.3 Results

We compare our $\beta$-divCSP method with three CSP baselines using different estimators for the covariance matrices. The first baseline uses the standard empirical estimator, the second one applies a standard analytic shrinkage estimator [9] and the third one relies on the minimum covariance determinant (MCDE) estimate [30]. Note that the shrinkage estimator usually provides better estimates in small-sample settings, whereas MCDE is robust to outliers. In order to perform a fair comparison we applied MCDE over various ranges $[0, 0.05, 0.1 \ldots 0.5]$ of parameters and selected the best one by cross-validation (as with $\beta$-divCSP). The MCDE parameter determines the expected proportion of artifacts in the data. The results are shown in Figure 3. Each circle denotes the error rate of one subject. One can see that the $\beta$-divCSP method outperforms the baselines as most circles are below the solid line. Furthermore the performance increases are significant according to the one-sided Wilcoxon sign rank test as the p-values are smaller than 0.05.

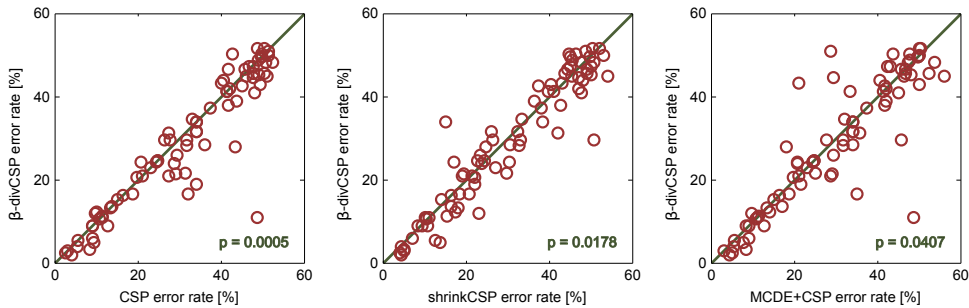

Figure 3: Performance results of the CSP, shrinkage + CSP and MCDE + CSP baselines compared to $\beta$-divCSP. Each circle represents the error rate of one subject. Our method outperforms the baselines for circles that are below the solid line. The p-values of the one-sided Wilcoxon sign rank test are shown in the lower right corner.

We made an interesting observation when analysing the subject with largest improvement over the CSP baseline; the error rates were 48.6% (CSP), 48.6% (MCDE+CSP) and 11.0% ($\beta$-divCSP). Over all ranges of MCDE parameters this subject has an error rate higher than 48% i.e. MCDE was not able help in this case. This example shows that $\beta$-divCSP and MCDE are not equivalent. Enforcing robustness on the CSP algorithm may in some cases be better than enforcing robustness when estimating the covariance matrices.

In the following we study the robustness property of the $\beta$-divCSP method on subject 74, the user with the largest improvement (CSP error rate: 48.6 % and $\beta$-divCSP error rate: 11.0 %). The left panel of Figure 4 displays the activity pattern associated with the most important CSP filter of subject 74. One can clearly see that the pattern does not encode neurophysiologically relevant activity, but focuses on a single electrode, namely FFC6. When analysing the (filtered) EEG signal of this electrode one can identify a strong artifact in one of the trials. Since neither the empirical covariance estimator nor the CSP algorithm is robust to this kind of outliers, it dominates the solution. However, the resulting pattern is meaningless as it does not capture motor imaginary related activity. The right panel of Figure 4 shows the relative importance of the divergence term of the artifactual trial with respect to the average divergence terms of the other trials. One can see that the divergence term computed from the artifactual trial is over 1800 times larger than the average of the other trials. This ratio decreases rapidly for larger $\beta$ values, thus the influence of the artifact decreases. Thus, our experiments provide an excellent example of the robustness property of the $\beta$-divCSP approach.

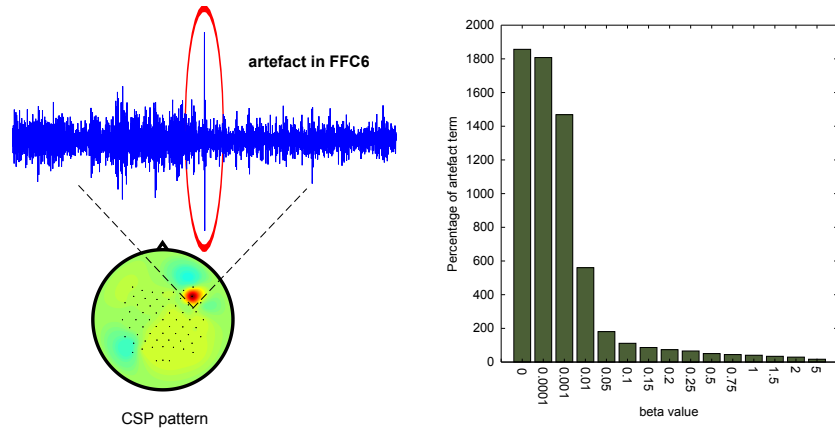

Figure 4: Left: The CSP pattern of subject 74 does not reflect neurophysiological activity but it represents the artifact (red ellipse) in electrode FFC6. Right: The relative importance of this artifactual trial decreases with the $\beta$ parameters. The relative importance is measured as quotient between the divergence term of the artifactual trial and the average divergence terms of the other trials.

## 5  Discussion

Analysis of EEG data is challenging because the signal of interest is typically present with a low signal to noise ratio. Moreover artifacts and non-stationarity require robust algorithms. This paper has placed its focus on a robust estimation and proposed a novel algorithm family giving rise to a beta divergence algorithm which allows robust spatial filter computation for BCI. In the very common setting where EEG electrodes become loose or movement related artifacts occur in some trials, it is a practical necessity to either ignore these trials (which reduces an already small sample size further) or to enforce intrinsic invariance to these disturbances into the learning procedures. Here, we have used CSP, the standard filtering technique in BCI, as a starting point and reformulated it in terms of an optimization problem maximizing the divergence between the class-distributions that correspond to two cognitive states. By borrowing the concept of beta divergences, we could adapt the optimization problem and arrive at a robust spatial filter computation based on CSP. We showed that our novel method can reduce the influence of artifacts in the data significantly and thus allows to robustly extract relevant filters for BCI applications.

In future work we will investigate the properties of other divergences for Brain-Computer Interfacing and consider also further applications like ERP-based BCIs [31] and beyond the neurosciences.

**Acknowledgment** We thank Daniel Bartz and Frank C. Meinecke for valuable discussions. This work was supported by the German Research Foundation (GRK 1589/1), by the Federal Ministry of Education and Research (BMBF) under the project Adaptive BCI (FKZ 01GQ1115) and by the Brain Korea 21 Plus Program through the National Research Foundation of Korea funded by the Ministry of Education.

## Appendix

**Sketch of proof of Theorem 1**
Cauchy's interlacing theorem [17] establishes a relation between the eigenvalues $\mu_1 \leq \ldots \leq \mu_D$ of the original covariance matrix $\boldsymbol{\Sigma}$ and the eigenvalues $\nu_1 \leq \ldots \leq \nu_d$ of the projected one $\mathbf{V}\boldsymbol{\Sigma}\mathbf{V}^\top$. The theorem says that

$$\mu_j \ \leq \ \nu_j \ \leq \ \mu_{D-d+j}.$$

In the proof we split the optimal projection $\mathbf{V}^*$ into two parts $\mathbf{U}_1 \in \mathbb{R}^{k \times D}$ and $\mathbf{U}_2 \in \mathbb{R}^{d-k \times D}$ based on whether the first or second trace term in Eq. (4) is larger when applying the spatial filters. By using Cauchy's theorem we then show that $\mathcal{L}_{kl}(\mathbf{U}) \leq \mathcal{L}_{kl}(\mathbf{W})$ where $\mathbf{W}$ consists of $k$ eigenvectors with largest eigenvalues; equality only holds if $\mathbf{U}$ and $\mathbf{W}$ coincide (up to linear transformations). We show an analogous relation for $\mathbf{U}_2$ and conclude that $\mathbf{V}^*$ must be the CSP solution (up to linear transformations). See the full proof in the supplement material.

**Sketch of the proof of Theorem 2**
Since there is only one discriminative direction we may perform analysis in a basis whereby the covariances of both classes have the form $\text{diag}(a, 1, \ldots, 1)$ and $\text{diag}(b, 1, \ldots, 1)$. If we show in this basis that consistency holds then it is a simple matter to prove consistency in the original basis. We want to show that as the number of trials $N$ increases the filter provided by sum$kl$-divCSP converges to the true solution $\mathbf{v}^*$. If the support of the density of the eigenvalues includes a region around $0$, then there is no hope of showing that the matrix inversion is stable. However, it has been shown in the random matrix theory literature [32] that if $D$ and $n$ tend to $\infty$ in a ratio $c = \frac{D}{n}$ then all of the eigenvalues apart from the largest lie between $(1 - \sqrt{c})^2$ and $(1 + \sqrt{c})^2$ whereas the largest sample eigenvalue ($\alpha$ denotes the true non-unit eigenvalue) converges almost surely to $\alpha + c\frac{\alpha}{\alpha-1}$ provided $\alpha > 1 + \sqrt{c}$, independently of the distribution of the data; a similar result applies if one true eigenvalue is smaller than the rest. This implies that for sufficient discriminability in the true distribution and sufficiently many data points per trial, each filter maximizing each term in the sum has non-zero dot-product with the true maximizing filter. But since the trials are independent, this implies that in the limit of $N$ trials the maximizing filter corresponds to the true filter. Note that the full proof goes well beyond the scope of this contribution.

## Footnotes

[1]The symmetric Kullback-Leibler Divergence between distributions $f(x)$ and $g(x)$ is defined as $\tilde{D}_{kl}(f(x) \, \| \, g(x)) = \int f(x) \cdot \log \frac{f(x)}{g(x)} dx \, + \, \int g(x) \cdot \log \frac{g(x)}{f(x)} dx$.

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
