[Supplementary Material]

# Robust Spatial Filtering with Beta Divergence

## Supplemental Material

**Wojciech Samek**[1,4]    **Duncan Blythe**[1,4]    **Klaus-Robert Müller**[1,2]    **Motoaki Kawanabe**[3]

[1]Machine Learning Group, Berlin Institute of Technology (TU Berlin), Berlin, German
[2]Department of Brain and Cognitive Engineering, Korea University, Seoul, Korea
[3]ATR Brain Information Communication Research Laboratory Group, Kyoto, Japan
[4]Bernstein Center for Computational Neuroscience, Berlin, Germany

## 1   Optimization Algorithm

The goal of our method is to find a projection $\mathbf{V} \in \mathbb{R}^{D \times d}$ to a subspace of dimensionality $d < D$ that maximizes a sum of divergences. Following [1] we decompose the projection into three parts, namely $\mathbf{V}^\top = \mathbf{I}_d \mathbf{R} \mathbf{P}$ where $\mathbf{I}_d$ is an identity matrix truncated to the first $d$ rows, $\mathbf{R}$ is a rotation matrix with $\mathbf{R}\mathbf{R}^\top = \mathbf{I}$ and $\mathbf{P}$ is the whitening matrix that projects the data onto an unit sphere. The optimization process then boils down to finding the rotation $\mathbf{R}$ that maximizes the sum of symmetric divergences

$$\mathcal{L}(\mathbf{R}) \;=\; \sum_i \tilde{D}\left((\mathbf{I}_d \mathbf{R} \mathbf{P})\boldsymbol{\Sigma}_1^i(\mathbf{P}^\top \mathbf{R}^\top \mathbf{I}_d^\top) \;\|\; (\mathbf{I}_d \mathbf{R} \mathbf{P})\boldsymbol{\Sigma}_2^i(\mathbf{P}^\top \mathbf{R}^\top \mathbf{I}_d^\top)\right).$$

Note that although $\mathbf{R}$ is a $D \times D$ rotation matrix, we only evaluate the first $d$ rows of it. The optimization is performed by gradient descend on the manifold of orthogonal matrices. More precisely, we start with an (random) orthogonal matrix $\mathbf{R}_0$ and find an orthogonal update $\mathbf{U}$ in the $k$-th step such that $\mathbf{R}_{k+1} = \mathbf{U}\mathbf{R}_k$. This way we stay on the manifold of orthogonal matrices at each step.

Note that the manifold of orthogonal matrices is connected to the set of skew-symmetric matrices $\mathbf{M} = -\mathbf{M}^\top$ via the exponential map [2]. Therefore we can express the orthogonal update matrix as $\mathbf{U} = e^{\mathbf{M}}$. The author of [3] provides a formula for the gradient of $f(\mathbf{U}) = f(e^{\mathbf{M}})$ at $\mathbf{U} = \mathbf{I} = e^{\mathbf{0}}$

$$\nabla_{\mathbf{M}} f(\mathbf{U})|_{\mathbf{M}=\mathbf{0}} = \left(\nabla_{\mathbf{U}} f(\mathbf{U})|_{\mathbf{U}=\mathbf{I}}\right)\mathbf{U}^\top - \mathbf{U}\left(\nabla_{\mathbf{U}} f(\mathbf{U})|_{\mathbf{U}=\mathbf{I}}\right)^\top.$$

With this we can determine the search direction $\mathbf{H} = -\mathbf{H}^\top$ in the set of skew symmetric matrices by computing the gradient of the loss function w.r.t. $\mathbf{M}$ at $\mathbf{M} = \mathbf{0}$. The update matrix can then be written as $\mathbf{U} = e^{t\mathbf{H}}$ where the optimal parameter $t$ is determined by performing a line-search.

Note that the divergence optimizes the whole subspace and the basis within the subspace is arbitrary. In order to extract uncorrelated sources[1] that maximally separate the two classes (as done by CSP), we select the principal axes of the data distribution of one class as basis .

## 2 Derivation of Kullback-Leibler Divergence CSP

The objective function of sum$kl$-divCSP (and $kl$-divCSP) can be written as

$$
\begin{aligned}
\mathcal{L}_{sumkl}(\mathbf{R}) &= \sum_i \tilde{D}_{kl}\left((\mathbf{I}_d\mathbf{RP})\boldsymbol{\Sigma}_1^i(\mathbf{P}^\top\mathbf{R}^\top\mathbf{I}_d^\top) \,\|\, (\mathbf{I}_d\mathbf{RP})\boldsymbol{\Sigma}_2^i(\mathbf{P}^\top\mathbf{R}^\top\mathbf{I}_d^\top)\right) \\
&= \frac{1}{2}\sum_i \left(\mathrm{tr}\left[((\mathbf{I}_d\mathbf{RP})\boldsymbol{\Sigma}_1^i(\mathbf{P}^\top\mathbf{R}^\top\mathbf{I}_d^\top))^{-1}((\mathbf{I}_d\mathbf{RP})\boldsymbol{\Sigma}_2^i(\mathbf{P}^\top\mathbf{R}^\top\mathbf{I}_d^\top))\right] + \right. \\
&\qquad \left. \mathrm{tr}\left[((\mathbf{I}_d\mathbf{RP})\boldsymbol{\Sigma}_2^i(\mathbf{P}^\top\mathbf{R}^\top\mathbf{I}_d^\top))^{-1}((\mathbf{I}_d\mathbf{RP})\boldsymbol{\Sigma}_1^i(\mathbf{P}^\top\mathbf{R}^\top\mathbf{I}_d^\top))\right] - 2d\right) \\
&= \frac{1}{2}\sum_i \left(\mathrm{tr}\left[(\bar{\boldsymbol{\Sigma}}_1^i)^{-1}\bar{\boldsymbol{\Sigma}}_2^i\right] + \mathrm{tr}\left[(\bar{\boldsymbol{\Sigma}}_2^i)^{-1}\bar{\boldsymbol{\Sigma}}_1^i\right] - 2d\right).
\end{aligned}
$$

Note that $\bar{\boldsymbol{\Sigma}}_1^i = (\mathbf{I}_d\mathbf{RP})\boldsymbol{\Sigma}_1^i(\mathbf{P}^\top\mathbf{R}^\top\mathbf{I}_d^\top)$ and $\bar{\boldsymbol{\Sigma}}_2^i = (\mathbf{I}_d\mathbf{RP})\boldsymbol{\Sigma}_2^i(\mathbf{P}^\top\mathbf{R}^\top\mathbf{I}_d^\top)$ denote the projected covariance matrices.

The gradient with respect to $\mathbf{R}$ can be computed as follows. Let us rewrite

$$
\nabla_{\mathbf{R}}\,\mathrm{tr}\left[((\mathbf{I}_d\mathbf{RP})\boldsymbol{\Sigma}_1^i(\mathbf{P}^\top\mathbf{R}^\top\mathbf{I}_d^\top))^{-1}((\mathbf{I}_d\mathbf{RP})\boldsymbol{\Sigma}_2^i(\mathbf{P}^\top\mathbf{R}^\top\mathbf{I}_d^\top))\right]
$$

as

$$
\mathbf{I}_d^\top\left[\nabla_{\mathbf{G}}\,\mathrm{tr}\left[\left(\mathbf{G}^\top\mathbf{C}\mathbf{G}\right)^{-1}\left(\mathbf{G}^\top\mathbf{D}\mathbf{G}\right)\right]\right]^\top
$$

with $\mathbf{G} = \tilde{\mathbf{R}}^\top$ and $\tilde{\mathbf{R}}$ is the $d \times D$ matrix consisting of the first $d$ rows of $\mathbf{R}$ and $\mathbf{C} = \mathbf{P}\boldsymbol{\Sigma}_1^i\mathbf{P}^\top$ and $\mathbf{D} = \mathbf{P}\boldsymbol{\Sigma}_2^i\mathbf{P}^\top$ are the whitened covariance matrices.

According to the matrix codebook [4] this gives

$$
\mathbf{I}_d^\top\left[-2\mathbf{C}\mathbf{G}(\mathbf{G}^\top\mathbf{C}\mathbf{G})^{-1}\mathbf{G}^\top\mathbf{D}\mathbf{G}(\mathbf{G}^\top\mathbf{C}\mathbf{G})^{-1} + 2\mathbf{D}\mathbf{G}(\mathbf{G}^\top\mathbf{D}\mathbf{G})^{-1}\right]^\top.
$$

Using this fact gives the following derivative $\nabla_{\mathbf{R}}\,\mathcal{L}_{sumkl}(\mathbf{R})$

$$
\mathbf{I}_d^\top\left(\sum_i (\bar{\boldsymbol{\Sigma}}_2^i)^{-1}\mathbf{I}_d\tilde{\boldsymbol{\Sigma}}_2^i - (\bar{\boldsymbol{\Sigma}}_1^i)^{-1}\bar{\boldsymbol{\Sigma}}_2^i(\bar{\boldsymbol{\Sigma}}_1^i)^{-1}\mathbf{I}_d\tilde{\boldsymbol{\Sigma}}_1^i + (\bar{\boldsymbol{\Sigma}}_1^i)^{-1}\mathbf{I}_d\tilde{\boldsymbol{\Sigma}}_1^i - (\bar{\boldsymbol{\Sigma}}_2^i)^{-1}\bar{\boldsymbol{\Sigma}}_1^i(\bar{\boldsymbol{\Sigma}}_2^i)^{-1}\mathbf{I}_d\tilde{\boldsymbol{\Sigma}}_2^i\right)\mathbf{R}
$$

where $\tilde{\boldsymbol{\Sigma}}_1^i = \mathbf{P}\boldsymbol{\Sigma}_1^i\mathbf{P}^\top$ and $\tilde{\boldsymbol{\Sigma}}_2^i = \mathbf{P}\boldsymbol{\Sigma}_2^i\mathbf{P}^\top$.

## 3 Derivation of Beta Divergence CSP

The objective function of $\beta$-divCSP can be written as

$$
\begin{aligned}
\mathcal{L}_\beta(\mathbf{R}) &= \sum_i \tilde{D}_\beta\left((\mathbf{I}_d\mathbf{RP})\boldsymbol{\Sigma}_1^i(\mathbf{P}^\top\mathbf{R}^\top\mathbf{I}_d^\top) \,\|\, (\mathbf{I}_d\mathbf{RP})\boldsymbol{\Sigma}_2^i(\mathbf{P}^\top\mathbf{R}^\top\mathbf{I}_d^\top)\right) \\
&= \frac{1}{\beta}\sum_i\left(\int g_i^{\beta+1}(x)dx + \int f_i^{\beta+1}(x)dx - \int f_i^\beta(x)g_i(x)dx - \int f_i(x)g_i^\beta(x)dx\right),
\end{aligned}
$$

with $f_i \sim \mathcal{N}\left(\mathbf{0}, \bar{\boldsymbol{\Sigma}}_1^i\right)$ and $g_i \sim \mathcal{N}\left(\mathbf{0}, \bar{\boldsymbol{\Sigma}}_2^i\right)$ being the zero-mean Gaussian distributions with projected covariances $\bar{\boldsymbol{\Sigma}}_1^i = (\mathbf{I}_d\mathbf{RP})\boldsymbol{\Sigma}_1^i(\mathbf{P}^\top\mathbf{R}^\top\mathbf{I}_d^\top) \in \mathbb{R}^{d\times d}$ and $\bar{\boldsymbol{\Sigma}}_2^i = (\mathbf{I}_d\mathbf{RP})\boldsymbol{\Sigma}_2^i(\mathbf{P}^\top\mathbf{R}^\top\mathbf{I}_d^\top) \in \mathbb{R}^{d\times d}$, respectively.

The integral $\int f_i^{\beta+1}(x)dx$ (and $\int g^{\beta+1}(x)dx$) can be expressed in explicit form as

$$
\begin{aligned}
\int f_i^{\beta+1}(x)dx &= \frac{1}{(2\pi)^{\frac{(\beta+1)d}{2}}|\bar{\boldsymbol{\Sigma}}_1^i|^{\frac{\beta+1}{2}}}\int e^{-\frac{1}{2}x^T(\frac{1}{\beta+1}\bar{\boldsymbol{\Sigma}}_1^i)^{-1}x}dx \\
&= \frac{1}{(2\pi)^{\frac{(\beta+1)d}{2}}|\bar{\boldsymbol{\Sigma}}_1^i|^{\frac{\beta+1}{2}}}(2\pi)^{\frac{d}{2}}\left(\frac{1}{\beta+1}\right)^{\frac{d}{2}}|\bar{\boldsymbol{\Sigma}}_1^i|^{\frac{1}{2}} \\
&= \frac{1}{(2\pi)^{\frac{\beta d}{2}}(\beta+1)^{\frac{d}{2}}}|\bar{\boldsymbol{\Sigma}}_1^i|^{-\frac{\beta}{2}}
\end{aligned}
$$

The integral $\int g_i^\beta(x)f_i(x)dx$ (and $\int f_i(x)g_i^\beta(x)dx$) can be expressed in explicit form as

$$
\begin{aligned}
\int g_i^\beta(x)f_i(x)dx &= \frac{1}{(2\pi)^{\frac{\beta d}{2}}|\bar{\Sigma}_2^i|^{\frac{\beta}{2}}}\frac{1}{(2\pi)^{\frac{d}{2}}|\bar{\Sigma}_1^i|^{\frac{1}{2}}}\int e^{-\frac{1}{2}x^T(\beta(\bar{\Sigma}_2^i)^{-1}+(\bar{\Sigma}_1^i)^{-1})x}dx \\
&= \frac{1}{(2\pi)^{\frac{\beta d}{2}}|\bar{\Sigma}_2^i|^{\frac{\beta}{2}}}\frac{1}{(2\pi)^{\frac{d}{2}}|\bar{\Sigma}_1^i|^{\frac{1}{2}}}(2\pi)^{\frac{d}{2}}\left|\beta(\bar{\Sigma}_2^i)^{-1}+(\bar{\Sigma}_1^i)^{-1}\right|^{-\frac{1}{2}} \\
&= \frac{1}{(2\pi)^{\frac{\beta d}{2}}|\bar{\Sigma}_2^i|^{\frac{\beta}{2}}|\bar{\Sigma}_1|^{\frac{1}{2}}}\left|\beta(\bar{\Sigma}_2^i)^{-1}+\Sigma_1^{-1}\right|^{-\frac{1}{2}} \\
&= \frac{1}{(2\pi)^{\frac{\beta d}{2}}}|\bar{\Sigma}_2^i|^{\frac{1-\beta}{2}}\left|\bar{\Sigma}_2^i(\beta(\bar{\Sigma}_2^i)^{-1}+(\bar{\Sigma}_1^i)^{-1})\bar{\Sigma}_1^i\right|^{-\frac{1}{2}} \\
&= \frac{1}{(2\pi)^{\frac{\beta d}{2}}}|\bar{\Sigma}_2^i|^{\frac{1-\beta}{2}}\left|\beta\bar{\Sigma}_1^i+\bar{\Sigma}_2^i\right|^{-\frac{1}{2}}
\end{aligned}
$$

Thus the objective function $\mathcal{L}_\beta(\mathbf{R})$ has the following explicit form

$$
\gamma\sum_i\left(|\bar{\Sigma}_1^i|^{-\frac{\beta}{2}}+|\bar{\Sigma}_2^i|^{-\frac{\beta}{2}}-(\beta+1)^{\frac{d}{2}}\left(|\bar{\Sigma}_2^i|^{\frac{1-\beta}{2}}|\beta\bar{\Sigma}_1^i+\bar{\Sigma}_2^i|^{-\frac{1}{2}}+|\bar{\Sigma}_1^i|^{\frac{1-\beta}{2}}|\beta\bar{\Sigma}_2+\bar{\Sigma}_1^i|^{-\frac{1}{2}}\right)\right),
$$

with $\gamma=\frac{1}{\beta}\sqrt{\frac{1}{(2\pi)^{\beta d}(\beta+1)^d}}$.

The gradient of $|\bar{\Sigma}_1^i|^{-\frac{\beta}{2}}$ with respect to $\mathbf{R}$ can be computed as

$$
\nabla_{\mathbf{R}}\left|(\mathbf{I}_d\mathbf{R}\mathbf{P})\Sigma_1^i(\mathbf{P}^\top\mathbf{R}^\top\mathbf{I}_d^\top)\right|^{-\frac{\beta}{2}}=\mathbf{I}_d^\top\left[\nabla_{\mathbf{G}}|\mathbf{G}^\top\mathbf{C}\mathbf{G}|^{-\frac{\beta}{2}}\right]^\top
$$

with $\mathbf{G}=\tilde{\mathbf{R}}^T$ and $\tilde{\mathbf{R}}$ is the $d\times D$ matrix consisting of the first $d$ rows of $\mathbf{R}$ and $\mathbf{C}=\mathbf{P}\Sigma_1^i\mathbf{P}^\top$. According to matrix codebook [4] this is

$$
-\beta\mathbf{I}_d^\top|\mathbf{G}^\top\mathbf{C}\mathbf{G}|^{-\frac{\beta}{2}}\cdot\left(\mathbf{C}\mathbf{G}(\mathbf{G}^\top\mathbf{C}\mathbf{G})^{-1}\right)^\top.
$$

Writing it back gives

$$
-\beta\mathbf{I}_d^\top|\bar{\Sigma}_1^i|^{-\frac{\beta}{2}}(\bar{\Sigma}_1^i)^{-1}\mathbf{I}_d\tilde{\Sigma}_1\mathbf{R}.
$$

where $\tilde{\Sigma}_1=\mathbf{P}\Sigma_1\mathbf{P}^\top$.

The gradient of the other term $|\bar{\Sigma}_2^i|^{\frac{1-\beta}{2}}|\beta\bar{\Sigma}_1^i+\bar{\Sigma}_2^i|^{-\frac{1}{2}}$ can be computed as

$$
\begin{aligned}
&\nabla_{\mathbf{R}}|(\mathbf{I}_d^\top\mathbf{R}\mathbf{P})\Sigma_2^i(\mathbf{P}^\top\mathbf{R}^\top\mathbf{I}_d^\top)|^{\frac{1-\beta}{2}}\cdot|\beta(\mathbf{I}_d\mathbf{R}\mathbf{P})\Sigma_1^i(\mathbf{P}^\top\mathbf{R}^\top\mathbf{I}_d^\top)+(\mathbf{I}_d\mathbf{R}\mathbf{P})\Sigma_2^i(\mathbf{P}^\top\mathbf{R}^\top\mathbf{I}_d^\top)|^{-\frac{1}{2}} \\
&=\mathbf{I}_d^\top\left[\nabla_{\mathbf{G}}\left(|\mathbf{G}^\top\mathbf{D}\mathbf{G}|^{\frac{1-\beta}{2}}\cdot|\beta\mathbf{G}^\top\mathbf{C}\mathbf{G}+\mathbf{G}^\top\mathbf{D}\mathbf{G}|^{-\frac{1}{2}}\right)\right]^T
\end{aligned}
$$

with $\mathbf{G}=\tilde{\mathbf{R}}^T$ and $\tilde{\mathbf{R}}$ is the $d\times D$ matrix consisting of the first $d$ rows of $\mathbf{R}$ and $\mathbf{C}=\mathbf{P}\Sigma_1^i\mathbf{P}^\top$ and $\mathbf{D}=\mathbf{P}\Sigma_2^i\mathbf{P}^\top$. According to the product rule this is

$$
-\mathbf{I}_d^\top\Big[(\beta-1)|\mathbf{G}^\top\mathbf{D}\mathbf{G}|^{-\frac{\beta+1}{2}}\cdot|\mathbf{G}^\top\mathbf{D}\mathbf{G}|\cdot\left(\mathbf{G}\mathbf{D}(\mathbf{G}^\top\mathbf{D}\mathbf{G})^{-1}\right)^\top\cdot|\beta\mathbf{G}^\top\mathbf{C}\mathbf{G}+\mathbf{G}^\top\mathbf{D}\mathbf{G}|^{-\frac{1}{2}}+
$$

$$
|\mathbf{G}^\top\mathbf{D}\mathbf{G}|^{\frac{1-\beta}{2}}\cdot|\mathbf{G}^\top(\beta\mathbf{C}+\mathbf{D})\mathbf{G}|^{-\frac{3}{2}}\cdot|\mathbf{G}^\top(\beta\mathbf{C}+\mathbf{D})\mathbf{G}|\cdot((\beta\mathbf{C}+\mathbf{D})\mathbf{G}(\mathbf{G}^\top(\beta\mathbf{C}+\mathbf{D})\mathbf{G})^{-1})^\top\Big]^\top
$$

Writing it back gives

$$
-\mathbf{I}_d^\top\Big((\beta-1)|\bar{\Sigma}_2^i|^{\frac{1-\beta}{2}}\cdot|\beta\bar{\Sigma}_1^i+\bar{\Sigma}_2^i|^{-\frac{1}{2}}\cdot(\bar{\Sigma}_2^i)^{-1}\mathbf{I}_d\tilde{\Sigma}_2^i+
$$

$$
|\bar{\Sigma}_2^i|^{\frac{1-\beta}{2}}\cdot|\beta\bar{\Sigma}_i+\bar{\Sigma}_2^i|^{-\frac{1}{2}}\cdot(\beta\bar{\Sigma}_1^i+\bar{\Sigma}_2^i)^{-1}\mathbf{I}_d(\beta\tilde{\Sigma}_1^i+\tilde{\Sigma}_2^i)\Big)^\top\mathbf{R}
$$

The total gradient is

$$
\nabla_{\mathbf{R}}\,\mathcal{L}_\beta(\mathbf{I}_d\mathbf{R}\mathbf{P}) \;=\; \mathbf{I}_d^\top\left(\gamma\sum_i -\beta|\bar{\boldsymbol{\Sigma}}_1^i|^{-\frac{\beta}{2}}(\bar{\boldsymbol{\Sigma}}_1^i)^{-1}\mathbf{I}_d\tilde{\boldsymbol{\Sigma}}_1^i \;-\; \beta|\bar{\boldsymbol{\Sigma}}_2^i|^{-\frac{\beta}{2}}(\bar{\boldsymbol{\Sigma}}_2^i)^{-1}\mathbf{I}_d\tilde{\boldsymbol{\Sigma}}_2^i\;+\right.
$$

$$
(\beta+1)^{\frac{d}{2}}|\bar{\boldsymbol{\Sigma}}_2^i|^{\frac{1-\beta}{2}}\cdot|\beta\bar{\boldsymbol{\Sigma}}_1^i\;+\;\bar{\boldsymbol{\Sigma}}_2^i|^{-\frac{1}{2}}\cdot\left[(\beta-1)(\bar{\boldsymbol{\Sigma}}_2)^{-1}\mathbf{I}_d\tilde{\boldsymbol{\Sigma}}_2^i\;+\;(\beta\bar{\boldsymbol{\Sigma}}_1^i\;+\;\bar{\boldsymbol{\Sigma}}_2^i)^{-1}\mathbf{I}_d(\beta\tilde{\boldsymbol{\Sigma}}_1^i\;+\;\tilde{\boldsymbol{\Sigma}}_2^i)\right]\;+
$$

$$
\left.(\beta+1)^{\frac{d}{2}}|\bar{\boldsymbol{\Sigma}}_1^i|^{\frac{1-\beta}{2}}\cdot|\beta\bar{\boldsymbol{\Sigma}}_2^i\;+\;\bar{\boldsymbol{\Sigma}}_1^i|^{-\frac{1}{2}}\cdot\left[(\beta-1)(\bar{\boldsymbol{\Sigma}}_1)^{-1}\mathbf{I}_d\tilde{\boldsymbol{\Sigma}}_1^i\;+\;(\beta\bar{\boldsymbol{\Sigma}}_2^i\;+\;\bar{\boldsymbol{\Sigma}}_1^i)^{-1}\mathbf{I}_d(\beta\tilde{\boldsymbol{\Sigma}}_2^i\;+\;\tilde{\boldsymbol{\Sigma}}_1^i)\right]\right)\mathbf{R}.
$$

## 4    Detailed Proof of Theorem 1

Note that [5] has provided a proof for the special case of one spatial filter. Let $\tilde{\mathbf{R}}\in\mathbb{R}^{d\times D}$ denote the orthogonal projection onto a subspace of dimension $d$ and let $\tilde{\boldsymbol{\Sigma}}_1$ and $\tilde{\boldsymbol{\Sigma}}_2$ represent the whitened covariance matrices with $\tilde{\boldsymbol{\Sigma}}_1+\tilde{\boldsymbol{\Sigma}}_2=\mathbf{I}$. Without loss of generality[2] we assume that $\tilde{\mathbf{R}}\tilde{\boldsymbol{\Sigma}}_1\tilde{\mathbf{R}}^\top=\boldsymbol{\Delta}_1$ and $\tilde{\mathbf{R}}\tilde{\boldsymbol{\Sigma}}_2\tilde{\mathbf{R}}^\top=\mathbf{I}-\boldsymbol{\Delta}_1$ with $\boldsymbol{\Delta}_1$ are diagonal matrices.

The KL divergence divCSP algorithm ($\lambda=0$) optimizes the following objective function $\mathcal{L}_{kl}(\tilde{\mathbf{R}})$ (ignoring constant terms)

$$
\mathrm{tr}\left((\tilde{\mathbf{R}}\tilde{\boldsymbol{\Sigma}}_1\tilde{\mathbf{R}}^\top)^{-1}(\tilde{\mathbf{R}}\tilde{\boldsymbol{\Sigma}}_2\tilde{\mathbf{R}}^\top)\right)\;+
$$

$$
\mathrm{tr}\left((\tilde{\mathbf{R}}\tilde{\boldsymbol{\Sigma}}_2\tilde{\mathbf{R}}^\top)^{-1}(\tilde{\mathbf{R}}\tilde{\boldsymbol{\Sigma}}_1\tilde{\mathbf{R}}^\top)\right)
$$

$$
=\;\;\mathrm{tr}\left(\boldsymbol{\Delta}_1^{-1}(\mathbf{I}-\boldsymbol{\Delta}_1)\right)+\mathrm{tr}\left((\mathbf{I}-\boldsymbol{\Delta}_1)^{-1}\boldsymbol{\Delta}_1\right)
$$

$$
=\;\;\sum_{i=1}^d\frac{1-\nu_i}{\nu_i}\;+\;\sum_{i=1}^d\frac{\nu_i}{1-\nu_i},
$$

where $\nu_i$ is the $i$-th diagonal element of $\boldsymbol{\Delta}_1$.

Let us decompose $\tilde{\mathbf{R}}=\begin{bmatrix}\mathbf{U}\\\mathbf{V}\end{bmatrix}$ into two matrices $\mathbf{U}\in\mathbb{R}^{k\times D}$ and $\mathbf{V}\in\mathbb{R}^{d-k\times D}$ as follows

$$
\mathbf{U}=\left\{\mathbf{r}_i:\frac{1-\nu_i}{\nu_i}>\frac{\nu_i}{1-\nu_i}\right\}\Longrightarrow\nu_i<0.5
$$

$$
\mathbf{V}=\left\{\mathbf{r}_i:\frac{1-\nu_i}{\nu_i}\le\frac{\nu_i}{1-\nu_i}\right\}\Longrightarrow\nu_i\ge0.5.
$$

Thus we can rewrite the objective function $\mathcal{L}_{kl}(\tilde{\mathbf{R}})$ as

$$
\underbrace{\sum_{i=1}^k\frac{1-\nu_i}{\nu_i}\;+\;\frac{\nu_i}{1-\nu_i}}_{\mathbf{U}}\;+\;\underbrace{\sum_{i=k+1}^d\frac{1-\nu_i}{\nu_i}\;+\;\frac{\nu_i}{1-\nu_i}}_{\mathbf{V}}\;.
$$

We prove that the top $d$ CSP filters $\mathbf{W}$, i.e. the top $d$ eigenvectors $\mathbf{v}_i$ ($i=1\ldots d$) of $\tilde{\boldsymbol{\Sigma}}_1$ sorted by $\alpha_i=\max\{\mu_i,1-\mu_i\}$ where $\mu_i$ denotes the $i$-th eigenvalue of $\tilde{\boldsymbol{\Sigma}}_1$, maximize $\mathcal{L}_{kl}(\tilde{\mathbf{R}})$. Let us divide $\mathbf{W}$ into $\tilde{\mathbf{U}}$ and $\tilde{\mathbf{V}}$ as done above.

Case 1: Assume $\tilde{\mathbf{R}}$ maximizes $\mathcal{L}_{kl}(\tilde{\mathbf{R}})$ and it consists of eigenvectors $\mathbf{v}_i$ of $\tilde{\boldsymbol{\Sigma}}_1$, but there exist $\mathbf{v}_j\in\tilde{\mathbf{R}}$ with $j>d$ (i.e. it is not among the top (according to the above sorting) $d$ eigenvectors). Thus $\mathbf{v}_j\notin\mathbf{W}$ and there exist $\mathbf{w}_l\in\mathbf{W}$ (which is among the top $d$ eigenvectors) with $\mathbf{w}_l\notin\tilde{\mathbf{R}}$.

Without loss of generality assume $\mathbf{v}_j\in\mathbf{U}$. In the following we prove

$$
\frac{1-\nu_j}{\nu_j}\;+\;\frac{\nu_j}{1-\nu_j}\;<\;\frac{1-\nu_l}{\nu_l}\;+\;\frac{\nu_l}{1-\nu_l},
$$

[2]Because the basis in the projected subspace is arbitrary, i.e. the Kullback-Leibler divergence is invariant to right multiplication of any non-singular matrix $\mathbf{G}\in\mathbb{R}^{d\times d}$ with $\mathcal{L}_{kl}(\mathbf{V})=\mathcal{L}_{kl}(\mathbf{V}\mathbf{G})$.

where $\nu_l$ and $\nu_j$ denote the diagonal element when applying $\mathbf{w}_l$ and $\mathbf{v}_j$, respectively. Note that the function $f(\nu) = \frac{1-\nu}{\nu} + \frac{\nu}{1-\nu}$ is maximized at the borders (one can show this by taking the derivative).

Assume $\mathbf{w}_l \in \tilde{\mathbf{U}}$. Then $\nu_l < \nu_j < 0.5$ because $\mathbf{w}_l$ is selected before $\mathbf{v}_j$ (remember $\mathbf{v}_j \notin \mathbf{W}$) according to above sorting. Thus $f(\nu_j) < f(\nu_l)$ as $f(\nu)$ is maximized for the smallest argument $\nu$ (if $\nu < 0.5$).

Assume $\mathbf{w}_l \in \tilde{\mathbf{V}}$. Then $1 - \nu_l < \nu_j < 0.5$ because $\mathbf{w}_l$ is selected before $\mathbf{v}_j$ according to above sorting. Thus $f(\nu_j) < f(1 - \nu_l) = f(\nu_l)$.

Let us define $\mathbf{B}$ as $\tilde{\mathbf{R}}$, but with $\mathbf{w}_l$ instead of $\mathbf{v}_j$. Thus $\mathcal{L}_{kl}(\tilde{\mathbf{R}}) < \mathcal{L}_{kl}(\mathbf{B})$. This is a contradiction to the assumption that $\tilde{\mathbf{R}}$ is the optimal solution.

Case 2: Assume $\tilde{\mathbf{R}}$ maximizes $\mathcal{L}_{kl}(\tilde{\mathbf{R}})$ and there exist (at least one) $\mathbf{r}_j \in \tilde{\mathbf{R}}$ with $\mathbf{r}_j$ is not an eigenvector of $\tilde{\boldsymbol{\Sigma}}_1$. Without loss of generality assume $\mathbf{r}_j \in \mathbf{U}$. Let us define a new solution $\mathbf{B} = \begin{bmatrix} \tilde{\mathbf{U}} \\ \tilde{\mathbf{V}} \end{bmatrix}$ as follows:

$\tilde{\mathbf{U}}$ consists of $k$ eigenvectors of $\tilde{\boldsymbol{\Sigma}}_1$ with smallest eigenvalues.
$\tilde{\mathbf{V}}$ consists of $d - k$ eigenvectors of $\tilde{\boldsymbol{\Sigma}}_1$ with largest eigenvalues.

Let us denote the diagonal elements (eigenvalues) of $\mathbf{U}\tilde{\boldsymbol{\Sigma}}_1\mathbf{U}^T$ as $\nu_1 < \ldots < \nu_k < 0.5$ and those obtained with $\tilde{\mathbf{U}}\tilde{\boldsymbol{\Sigma}}_1\tilde{\mathbf{U}}^T$ as $u_1 < \ldots < u_k < 0.5$. Note that $u_i = \mu_i$ where $\mu_1 < \ldots < \mu_D$ are the eigenvectors of $\tilde{\boldsymbol{\Sigma}}_1$ (because $\tilde{\mathbf{U}}$ consists of the smallest eigenvectors of $\tilde{\boldsymbol{\Sigma}}_1$). Cauchy's interlacing theorem [6] establishes the following relation between $\nu_i$ and $u_i$, namely $u_i \leq \nu_i$. Note that equality only holds if $\mathbf{U}$ and $\tilde{\mathbf{U}}$ are the same, i.e. if $\mathbf{U}$ consists of the eigenvectors of $\tilde{\boldsymbol{\Sigma}}_1$ (irrespectively of permutation). Cauchy's theorem implies that there are no $\nu_i$ and $\nu_j$ with $u_k < \nu_i < \nu_j < u_{k+1}$. Together with the fact that $f(\nu) = \frac{1-\nu}{\nu} + \frac{\nu}{1-\nu}$ is maximized at the borders (i.e. for smallest $\nu$ in this case) this for all $i$ implies

$$\frac{1 - \nu_i}{\nu_i} + \frac{\nu_i}{1 - \nu_i} \leq \frac{1 - u_i}{u_i} + \frac{u_i}{1 - u_i},$$

Since $\exists i$ where this relation is strictly positive (because we assumed $\mathbf{r}_j \in \mathbf{U}$), we obtain $\mathcal{L}_{kl}(\mathbf{U}) < \mathcal{L}_{kl}(\tilde{\mathbf{U}})$.

Let us denote the diagonal elements (eigenvalues) of $\mathbf{V}\tilde{\boldsymbol{\Sigma}}_1\mathbf{V}^T$ as $\nu_1 > \ldots > \nu_{d-k} \geq 0.5$ and those obtained with $\tilde{\mathbf{V}}\tilde{\boldsymbol{\Sigma}}_1\tilde{\mathbf{V}}^T$ as $u_1 > \ldots > u_{d-k} \geq 0.5$. Note that $u_i = \mu_i$ where $\mu_1 > \ldots > \mu_D$ are the eigenvectors of $\tilde{\boldsymbol{\Sigma}}_1$ (because $\tilde{\mathbf{V}}$ consists of the largest eigenvectors of $\tilde{\boldsymbol{\Sigma}}_1$). Cauchy's interlacing theorem establishes the following relation between the $\nu_i$ and $u_i$, namely $\nu_i \leq u_i$. Note that equality only holds if $\mathbf{V}$ and $\tilde{\mathbf{V}}$ are the same (irrespectively of permutation). Together with the fact that $f(\nu) = \frac{1-\nu}{\nu} + \frac{\nu}{1-\nu}$ is maximized at the borders (i.e. for largest $\nu$ in this case) this implies

$$\frac{1 - \nu_i}{\nu_i} + \frac{\nu_i}{1 - \nu_i} \leq \frac{1 - u_i}{u_i} + \frac{u_i}{1 - u_i},$$

Thus $\mathcal{L}_{kl}(\mathbf{V}) \leq \mathcal{L}_{kl}(\tilde{\mathbf{V}})$ and consequently $\mathcal{L}_{kl}(\tilde{\mathbf{R}}) = \mathcal{L}_{kl}(\tilde{\mathbf{U}}) + \mathcal{L}_{kl}(\tilde{\mathbf{V}}) < \mathcal{L}_{kl}(\tilde{\mathbf{U}}) + \mathcal{L}_{kl}(\tilde{\mathbf{V}}) = \mathcal{L}_{kl}(\tilde{\mathbf{B}})$.

This contradicts the assumption that $\tilde{\mathbf{R}}$ maximizes $\mathcal{L}_{kl}(\tilde{\mathbf{R}})$.

## Footnotes

[1]Spatial filters $\mathbf{v}_i$ and $\mathbf{v}_j$ $(i \neq j)$ extract uncorrelated source $\mathbf{s}$ as $\mathbf{v}_i^\top \boldsymbol{\Sigma} \mathbf{v}_j = \mathbf{v}_i^\top \mathbf{X}\mathbf{X}^\top \mathbf{v}_j = \mathbf{s}^\top \mathbf{s} = 0$.