[Reviews · NeurIPS 2013]

Submitted by Assigned_Reviewer_1

The authors present a new approach to designing robust common spatial pattern (CSP) filters in the context of brain-computer interfacing (BCI). They evaluate their algorithm on experimental data from 80 subjects and compare it favourably with ordinary CSP and CSP with regularization of the covariance matrices as proposed by Ledoit-Wolf.

I find the idea of this work intriguing and the manuscript very well written. The math is elegant and the experimental results are quite good. I have two (closely related) issues though that really bother me:

* Regarding spatial filtering for BCIs, the authors almost exclusively cite work of the BCI group working at TU-Berlin. I find this very unprofessional. There are plenty of groups that have published relevant work on this topic, at NIPS as well as in high-quality journals, and these groups deserve credit for their work.
* The authors limit the comparison of their algorithm to vanilla CSP and the Ledoit-Wolf regularization. I doubt their algorithm would also compare favourably to state-of-the-art algorithms developed by other groups.
Summary: A very intriguing idea, very well-written manuscript, and good experimental results. However, the authors mostly ignore the work of groups other than the one at TU-Berlin and do not compare the performance of their algorithm with similarly sophisticated algorithms developed by other researchers.

Submitted by Assigned_Reviewer_4

The contribution presents a new perspective on CSP computation using
Beta divergences showing that increasing Beta yields more robust
estimates and better classification performance of simulations
and real EEG data from 80 subjects.

The paper is well written and the proposed method is mathematically
elegant although I have some major comments.

- The authors compare their method to the Ledoit-Wolf estimation
claiming that the L2 shrinkage it provides in closed form addresses
the problem of outliers, although the purpose is to reduce bias
when the number of samples is small. Their exists proper
robust covariance estimation methods (M-estimator, Minimum Covariance
Determinant [Rouseeuw1999]) that would be much better fits for
fair comparison.

- Results provided in Figure 3, although statistically significant
across the 80 subjects, demonstrate limited improvement on average
with a large increase in computation time.

Typos:
- top of page 3 'the the true solution'
- ref 17 has a problem
- ref 21 type EEG in capital letters
Summary: Paper is mathematically elegant, explaining how beta divergences
can be used for down-weigthing outliers in CSP computations.
The performance on real data seem however limited, and a fair
comparison with dedicated robust covariance estimators would
have made the contribution stronger.

Submitted by Assigned_Reviewer_5

This paper shows that the CSP algorithm (a very common algorithm for spatial filtering EEG data to increase the discriminability between classes characterized by differences in power/variance as in motor imagery) can be understood as a divergence maximization algorithm. They then use that insight to develop a CSP algorithm that is more robust to outliers by using the beta divergence.

Clarity

The paper is very clearly written. Figure 1 is a very nice summary of the early parts of the paper.

Quality

This paper is a very nice example of a theoretical contribution that ends up having practical application to real data. One minor concern I had was that I am not sure how realistic adding independent noise to each electrode is as many EEG artifacts are visible across multiple electrodes (e.g. blinks, muscle movements, EM noise, physical movements), however the authors do show an example of exactly this type of artifact in their real data. It would be nice to see how the algorithm performs on other types of artifacts and non-stationarities.

Significance

Improving the CSP algorithm could have a big impact on motor-imagery and steady-state evoked potential BCIs.

Originality

The authors mention that the proof that CSPs maximize divergence has been proved for the special case of 1 CSP filter. The rest of the work appears to be novel.
Summary: This paper is a very nice example of a theoretical contribution that ends up having practical application to real data. It is clearly written, novel, and could have large impact.
Author Feedback

Author rebuttal: We thank the reviewers for their efforts and their helpful comments.

In the following we address the four concerns mentioned by the reviewers:

1. Focussing on the work of the TU-Berlin BCI group (Reviewer 1).

We fully agree with the reviewer that many groups contributed to the field of BCI/CSP and deserve credit for their work. It has not been our intention to focus exclusively on the work performed by the TU-Berlin group.
Please note that we also gave credit to other authors working in the field of BCI/CSP e.g. [2] Wolpaw et al., [4] Ramoser et al., [6] Lotte and Guan, [12] Arvaneh et al. and [21] Wang.
The reason why the work of the TU-Berlin group may be slightly over-represented in our citations is simply the fact that we use their optimization framework to maximize our objective function.
In order to avoid such over-representation, we will add the following citations to the revised manuscript: Lu et al. 2010, Devlaminck et al. 2011, Parra et al. 2005, Yong et al. 2008. Furthermore we will delete [5] and [10] from the references.
----

2. Comparison to vanilla CSP and Ledoit-Wolf regularization (Reviewer 1+2).

We thank the reviewers for poining out that the Ledoit-Wolf estimation may not be the most appropriate method to compare against. We are aware that this approach was especially designed for small-sample settings. Note that this kind of regularization approach was applied in [6] Lotte and Guan, thus can be regarded as one of the state-of-the-art CSP algorithms developed by other researchers.
We apply the Ledoit-Wolf covariance estimator to each trial separately in order to improve its estimation quality. Since the problem of trial-wise covariance estimation is to some extend a small-sample problem we think that this kind of comparison is reasonable. We also tried to apply shrinkage with cross-validated shrinkage parameter (same over all trials), but the performance was worse than the analytical shrinkage that allows different shrinkage strength over trials.

We thank reviewer 2 for pointing us at the work of [Rouseeuw1999]. We agree that comparison with a robust covariance estimation method is very reasonable.
Using the freely available Matlab code we applied MCDE [Rouseeuw1999] and estimated the covariance matrices from the pooled data. Note that only the application of the algorithm to the whole data set allows to downweight outlier trials (it also gave better results than when applying MCDE to each trial separately). This kind of approach (MCDE+CSP) has been proposed by Yong et al. 2008 thus it can also be seen as one of the state-of-the-art CSP methods.
In order to perform a fair comparison we applied MCDE over various ranges of parameters and selected the best one by cross-validation (as with beta-divCSP). The MCDE parameter determines the expected proportion of artifacts in the data. The beta-divCSP method significantly (p = 0.0407) outperforms MCDE+CSP, the mean (median) error rates are 29.3 (29.7) and 30.2 (31.7), respectively.

We made an interesting observation when analysing the subject with largest improvement, his error rates were 48.6 (CSP), 48.6 (MCDE+CSP) and 11.0 (beta-divCSP).
Over all ranges of MCDE parameters this subject has an error rate higher than 48 % i.e. MCDE was not able help in this case.
This example show that beta-divCSP and MCDE are not equivalent.
Enforcing robustness on the CSP algorithm may in some cases be better than enforcing robustness when estimating the covariances.
We think that the pros and cons of both robustness approaches should be studied in future research as they may be highly relevant for practical applications.
We will of course add all these results and discussion to the revised version of the manuscript.

As a final remark we would like to note that although there are many sophisticated CSP variants, many of them use information from other subjects (Devlaminck 2011, Lotte 2010, Kang 2009) or other sessions (Blankertz 2008, Bamdadian 2008), use adaptation (Vidaurre 2011, Shenoy 2006) or do not focus on robustness but rather on stationarity (Samek 2012, Arvaneh 2013).
We do not compare our method with these algorithms as they either solve a different problem or use additional information.
We think that comparing our approach to shrinkage+CSP and to MCDE+CSP is appropriate.
----

3. Limited average improvement (Reviewer 2)

We agree with the reviewer that our method does not provide large performance improvement for all subject. However, we think that this is the case for (almost) all new methods introduced into the field of BCI. Our method is helpful if large artifacts are present in the data, it does not necessary improve performance when this is not the case.
Please note that beta-divCSP is able to improve classification accuracy for all ranges of performance. In other words our method does not only help subjects with error rates far above 30%, but it is also able to improve performance for good subjects (see Figure 3).

Furthermore please note that we did not only contribute yet another CSP variant, but also provide (and proof) a novel view on CSP. The connection between divergences and CSP allows to apply results from the field of information geometry to the CSP problem. Beta divergence is only one kind of divergence which is known, other divergences may also be applicable and have other interesting properties.
----

4. Adding noise to only one electrode in the simulations (Reviewer 3).

Please note that we do not add noise to only one electrode in section 4.1. We add noise to every electrode and trial with a certain probability. This allows to model artifacts over multiple electrodes as suggested by the reviewer. A proper study on the impact of different classes of artifacts (with more advanced artifact models) on the (beta-div)CSP algorithm would be very interesting, unfortunately, it is out of scope of this paper.